# Current Insights into the Significance of the Renal Resistive Index in Kidney and Cardiovascular Disease

**DOI:** 10.3390/diagnostics13101687

**Published:** 2023-05-10

**Authors:** Roxana Darabont, Diana Mihalcea, Dragos Vinereanu

**Affiliations:** 1Cardiology and Cardiovascular Surgery Department, University of Medicine and Pharmacy Carol Davila, 050098 Bucharest, Romania; rdarabont@yahoo.com (R.D.); vinereanu@gmail.com (D.V.); 2Cardiology Department, University and Emergency Hospital, 050098 Bucharest, Romania

**Keywords:** renal resistive index, acute and chronic kidney disease, arterial stiffness, cardiovascular disease

## Abstract

Initially, the renal resistive index (RRI) was investigated with the aim of improving diagnosis in kidney diseases, but this goal was not met. Recently, many papers have highlighted the prognostic significance of the RRI in chronic kidney disease: specifically, in estimating the revascularization success of renal artery stenoses or the evolution of the graft and the recipients in renal transplantation. Moreover, the RRI has become significant in the prediction of acute kidney injury in critically ill patients. Studies in renal pathology have revealed correlations of this index with parameters of systemic circulation. The theoretical and experimental premises of this connection were then reconsidered, and studies analyzing the link between RRI and arterial stiffness, central and peripheral pressure, and left ventricular flow were conducted with this purpose. Many data currently indicate that RRI is influenced more by pulse pressure and vascular compliance than by renal vascular resistance—assuming that RRI reflects the complex interplay between systemic circulation and renal microcirculation and should be considered a marker of systemic cardiovascular risk beyond its prognostic relevance for kidney disease. In this review, we overview the clinical research that reveals the implications of RRI in renal and cardiovascular disease.

## 1. Introduction

The resistive index was defined by Léandre Pourcelot in 1974 as the difference between peak systolic and end-diastolic velocity divided by the peak systolic velocity, as measured using an arterial ultrasound Doppler waveform [1]. As its name suggests, it was initially considered a parameter of vascular resistance in different territories. The renal resistive index (RRI) was initially investigated with the aim of improving diagnosis in urinary obstruction [2] or renal vein thrombosis [3,4]. Due to its lack of specificity, the RRI was soon shown to be unable to contribute to the specific diagnosis of different renal pathological conditions. Despite this failure, the RRI continued to be studied. In the last 30 years, many research works have sought to identify the prognostic significance of RRI in chronic kidney disease (CKD), in the prediction of revascularization success in renal artery stenoses (RAS), and in the evolution of grafts and recipients in renal transplantation. Moreover, in recent years, consistent data have accumulated regarding the importance of RRI in the prediction of acute kidney injury (AKI) in different categories of critically ill patients.

Additionally, studies in renal pathology have revealed the correlations of the RRI with parameters of systemic circulation. The theoretical and experimental premises of this connection were reviewed, and studies dedicated to analyzing the link between RRI and arterial stiffness, central and peripheral pressure, and left ventricular flow were designed specifically for these objectives. Many data currently indicate that RRI is much more heavily influenced by the pulse pressure and vascular compliance than by renal vascular resistance—facts that allow us to assume that the RRI primarily reflects the complex interplay between systemic circulation and renal microcirculation, and that it should be considered as a marker of systemic cardiovascular risk beyond its prognostic significance for kidney disease. In this review, we offer an overview of the clinical research that has revealed the most important implications of RRI for renal and cardiovascular disease in the last few decades.

## 2. Evaluation of the Renal Resistive Index

Duplex ultrasound involves visualizing the anatomical landmarks of a certain region of interest in B mode, identifying the vessels based on the color Doppler application, and recording the blood flow parameters through a spectral signal. Thus, this type of examination has the advantage of providing both morphological and functional information by using a non-invasive, low-cost, and highly sensitive technology [5,6,7]. Table 1 presents the equation that defines the RRI and the standardized requirements for its correct measurement.

Conditions that could influence the accuracy of RRI measurements include severe hypotension, arrhythmias, the Valsalva maneuver, and renal compression due to perirenal or subcapsular fluid collections [10].

Figure 1 illustrates how RRI is assessed using a Doppler ultrasound waveform at the level of a renal interlobar artery.

## 3. Determinants of the Renal Resistive Index

Nowadays, the RRI is no longer perceived only as a measure of renal vascular resistance, but rather as a parameter that reflects the hemodynamic conditions both of the renal microcirculation and those of the systemic circulation, in variable proportions imposed by certain pathological contexts.

The first category of arguments in favor of this statement are theoretical. Charles O’Neill has shown that, by rearranging the common equation of RRI and by replacing the velocities with the ratio between pressure gradient (ΔP) and the product of resistance with the lumen area (R × LA), it transpires that RRI does not depend on vascular resistance:$$\mathrm{RRI}\;=\;1\;-\;\frac{Pdiast-P_{0}}{Psyst-P_{0}}\times \frac{LA\;systolic}{LA\;diastolic},$$
where P_0_ represents a combination of interstitial pressure and venous pressure (renal capillary wedge pressure), the fraction of pressures is a function of the pulse pressure, and the fraction of lumen areas in systole and diastole is a function of compliance.

This type of calculation clearly indicates that the RRI varies with interstitial and capillary pressure in the kidney (P_0_); it is directly associated with the pulse pressure (PP) and inversely associated with the vascular compliance [11].

In line with these theoretical premises, experimental studies emphasized the hemodynamic circumstances in which RRI varies with vascular compliance and vascular resistance. To better understand the results of these studies, we must remember that the intrarenal flow is determined by two opposite parameters: the pressure gradient between the aorta and intrarenal arteries and the intra-renal vascular resistance. Compliance represents the rate of change in the volume of a vessel as a function of pressure. In vitro experiments have shown that the RRI becomes less dependent on resistance as compliance decreases, being completely independent of vascular resistance when compliance is zero [12]. These results were confirmed by a series of ex vivo experiments. Although a linear relationship was found between the RRI and pharmacologically induced changes in vascular resistance, RRI increased also in association with vascular resistance that were likely non-physiologic. On the contrary, the RRI was markedly changed while PP is increasing [13]. In addition to these data, the expansive experiments of Albany group and those of Claudon et al. have proved that an increase in interstitial pressure caused by ureteral obstruction can lead to a drastic reduction in the cross-sectional area of renal arterioles in diastole and, as such, to systolic velocities higher than the diastolic velocities and to an augmented RRI [14,15].

It is also worth discussing the evidence from clinical trials supporting the strong relationship between systemic and intrarenal circulation, which is mostly reflected in the variations in RRI with arterial stiffness, central and peripheric PP, and diastolic blood pressure (DBP). The best illustration of this correlation concerns the fact that the RRI in transplanted kidneys varies with the age, general prognosis, and systemic vascular parameters of the recipient rather with the histopathological characteristics and survival of the graft [16,17].

Aortic stiffness exerts an influence on RRI through the augmentation of systolic central pressure and decreases in diastolic central pressure. Consequently, PP is significantly amplified. Like the brain, the kidneys are vulnerable to high-pressure fluctuations that may multiply three to four times in amplitude with advancing age. Furthermore, pulsatile stress can induce endothelial dysfunction in small renal arteries [18]. On the other hand, microvascular kidney disease can also manifest itself, concomitantly or not, with stiffening of the central arteries, leading to decreased compliance and high RRI values. This assumption is supported by biopsy studies showing that, from a range of histological abnormalities, only renal arteriosclerosis independently correlates with RRI [19,20]. In this context, it can be difficult to differentiate the contribution of each vascular territory to the rise of the RRI and to renal dysfunction.

There are two more hemodynamic factors that can increase the RRI: high renal blood flow [21] and a slow heart rate. Bradycardia increases the RRI because a prolonged diastole decreases more the end-diastolic velocity [22]. Some authors have directly recorded the inverse correlation of the RRI with heart rate [17,23], while others have deduced it from the correlation of RRI values with use of beta-blockers [24,25]. Regarding the anthropometric determinants, it has been shown that RRI increases with age [8,23,24,25,26,27,28,29] and body mass index [8,24,28], is slightly higher in females [8,24,25], and is inversely correlated with body height [23,24].

Recently, a study conducted in the general population in Switzerland showed an association between the RRI and sodium intake, estimated based on 24 h urine samples. The authors hypothesized that the impact of salt on renal hemodynamics may be due either to functional and structural changes in the intra-renal vessels, or to an inadequate renal vasomotor response [30]. The main determinants of the RRI are reviewed in Figure 2.

### 3.1. Renal Resistive Index in Kidney Disease

**Chronic kidney disease.** The RRI was initially and extensively studied in relation to renal pathology. The first studies focused on its diagnostic importance. Pioneering studies drew attention to significantly increased RRI values in kidneys affected by urinary tract obstruction [2,31]. It was later shown that the sensitivity of the RRI to differentiate between the two kidneys (ΔRRI) was too low, even for the diagnosis of complete urinary obstruction [32]. In turn, similar expectations in the diagnosis of renal vein thrombosis were subsequently refuted [3,33]. An evaluation of the RRI in relation to renal biopsies revealed its inability to distinguish between various forms of renal parenchymal diseases [34]. Although the RRI can increase in different types of renal lesions [35], it is currently thought that, of all histological anomalies, the RRI correlates best with renal arteriolosclerosis [18,19].

An impressive number of data currently support the prognostic value of RRI in CKD. This information comes from studies with variable designs. First, there are criteria for CKD diagnosis, based either on biological [24,36,37,38,39,40,41] or on morphological parameters, provided by biopsy examinations [42]. Second, the RRI values from which its increase begins to be correlated with the degradation of renal function are between 0.65 [40] and 0.80 [36], with most studies applying the delimitation to 0.70 [36,38,39,42]. Third, the definition for worsening renal function included different amounts of serum creatinine variations [36,37,41] or a decrease in the estimated glomerular filtration rate (eGFR) from the baseline with 5 mL/min/1.73 m^2^/year [40] or with at least 20 mL/min/1.73 m^2^/ > 50% until the end of the follow-up period [38,39,41,42]. All of the above-mentioned criteria were analyzed in conjunction with the occurrence of end-stage renal disease (ESRD) with the need for replacement therapy. Moreover, the duration of the follow-up period ranged from two [38] to six years [42]. Despite their lack of homogeneity, all these studies concluded that an increased RRI, along with proteinuria, low eGRF at baseline, and hypertension, is an independent risk factor for worsening renal dysfunction.

Two studies, however, contradict this evidence. One of these works looked retrospectively at 131 patients with non-proteinuric CKD for a period of 7.5 years. Their results indicated that patients with an RRI ≥ 0.80 have a faster increase in serum creatinine compared with those with an RRI < 0.80 at baseline, and each 0.1 increment of RRI was an independent determinant of 5-year renal disease progression and a predictor of mortality. However, as a single marker, the RRI showed poor discrimination performance [43]. The other study is a prospective study, part of the project titled Cardiovascular and Renal Outcomes in CKD 2–4 Patients—The Fourth Homburg Evaluation (CARE FOR HOMe). During the external validation of the kidney failure risk equation, which includes age, gender, eGFR, and the urinary albumin-to-creatinine ratio [44], routine duplex examination did not improve risk prediction for ESRD [45]. The main studies that have evaluated the prognostic significance of RRI in CKD are summarized in Table 2.

**Diabetic kidney disease.** The RRI proved to be significantly higher in patients with diabetic kidney disease than in those with different chronic renal diseases [46,47,48]. In diabetic kidney disease, the RRI increases with the severity or progression of renal disease, as is the case in any other CKD [49,50,51,52,53]. In a study of 157 hypertensive patients with diabetes mellitus and microalbuminuria, who were followed for 7.8 years, a decrease ≥3 mL/min/1.73 m^2^/year for eGFR was encountered 2 to 3 times more frequently in those with a RRI ≥ 0.80, while regression to albuminuria was seen less frequently in this category of patients compared with those with RRI < 0.80 [54]. The most promising investigations were carried out on the potential role of RRI in the differential diagnosis of diabetic kidney disease. A series of 469 type 2 diabetes patients who underwent renal biopsies was consecutively reviewed. The RRI was significantly higher in the diabetic kidney disease group compared to those without it. The optimum cut-off value of RRI for predicting diabetic kidney disease was 0.66 and was proposed to be integrated in a prediction model along with HbA1c ≥7%, diabetes duration ≥ 60 months, diabetic retinopathy, and the body mass index [55]. Other authors have suggested that RRI values > 0.72 may be in favor of diabetic glomerulosclerosis compared with renal lesions with another substrate in type 2-diabetic patients [56].

**Renal artery stenosis.** Despite some previous reports to the contrary, the RRI’s contribution to diagnostic approaches to renal artery stenosis is considered limited, even in the case of critical stenosis (>80%) [57,58]. This goal might be better achieved by a combined parameter obtained by subtracting the splenic resistive index from the RRI, which has significantly lower values (−0.05 vs. 0.068) in the presence of a renal artery stenosis, but more data are needed to confirm this hypothesis [59].

The RRI was studied with great interest in relation to its ability to predict revascularization success in renovascular disease. Starting with the work of Radermacher et al., it has been emphasized that a RRI ≥ 0.80 could be a predictor of no improvement in blood pressure, renal function, and kidney survival [60]. Soulez et al. went further and analyzed the predictive values of RRI before and after captopril administration or in conjunction with kidney length [61]. However, as some authors have remarked, RRI is influenced by too many hemodynamic factors to be a reliable determinant of the success of renal revascularization. For example, a low intrarenal post-stenotic RRI may indicate a stenosis of increased severity, which is more likely to respond to intervention than a low- or moderate-grade stenosis [62]. Additionally, it remains an open question as to whether RRI in the contralateral kidney would not be a better predictor of renal outcomes after an intervention for a unilateral renal artery stenosis [63].

**Renal transplant.** Initially, there were several encouraging results related to the RRI’s prognostic significance in graft evolution [64,65,66]. Subsequently, most studies reached a similar conclusion: the RRI is not able to differentiate between the medical complications of an allograft, and the causal diagnosis of graft dysfunction should be performed only by biopsy [67]. However, when analysis was extended to the RRI in relation to recipient characteristics, it was found that an increased RRI in the graft is associated with recipient mortality [16,68,69,70,71], age [15,16], pulse pressure, or parameters of arterial stiffness [15,16,69,70,71]—data that indicate the significant influence of systemic circulation on the RRI.

**Acute kidney injury.** The RRI was analyzed in relation to AKI from two main perspectives: its usefulness in distinguishing between reversible (pre-renal) and persistent (acute tubular necrosis) renal injury and the prediction of the development of AKI in critically ill patients [72].

When it comes to differential diagnosis between reversible and irreversible AKI, several studies and a meta-analysis supporting the notion that irreversible AKI is characterized by higher RRI values, the differentiation in relation to the reversible forms of AKI being above 0.75 [73,74,75,76]. Once again, a retrospective study disproves the ability of the RRI to predict persistent AKI in patients with septic shock because it did not improve a prediction model based on a combination of serum creatinine and the non-renal SOFA score [77].

The ability of RRI to predict AKI was studied in different clinical settings: in critically ill patients [78], in shock [79,80], in cardiac surgery with cardio-pulmonary bypass [81], and after TAVR [82]. RRI values ranging from 0.70 to 0.795 proved to be significantly correlated with the risk of AKI in these categories of patients [79,80,81,82,83]. The main studies referring to the implications of RRI in AKI are listed in Table 3.

**Evolution of the renal resistive index under therapy.** Few studies have studied how the RRI varies under different therapeutic agents. In a study that included a relatively small number of hypertensive patients, lisinopril was associated with a significant decrease in the urinary albumin/creatinine ratio and in the RRI compared with nifedipine GITS [83]. More recently, a pilot study of diabetic patients showed that the RRI significantly decreases during 2 days of treatment with dapagliflozin 10 mg/day, along with the pulse wave velocity and endothelial dysfunction, evaluated using flow-mediated dilatation [84]. No less importantly, improvements in the RRI have also been identified after catheter-based renal sympathetic denervation in patients with resistant hypertension [85].

### 3.2. Renal Resistive Index and the Cardiovascular System

The RRI assesses renal microcirculation in response to several pathologies [86]. In addition, various hemodynamic renal and extrarenal factors influence RRI quantification [86]. Among the renal factors, the most important are capillary wedge pressure and interstitial and venous pressure, while arterial vascular compliance, cardiac function, and systolic and diastolic blood pressure have significant extrarenal value [86]. Moreover, heart rate variability influences RRI values independently from other hemodynamic parameters [28]. If bradycardia determines a high RR through an increased diastolic flow, tachycardia favors a decreased RRI due to the shortening of the diastole [28,87].

**Arterial stiffness**, a marker of macrovascular disease, is an independent predictor of cardiovascular morbidity and mortality in patients with diabetes, hypertension, dyslipidemia, and renal insufficiency [28]. The RRI, a parameter of microvascular pathology, evaluates renal vascular resistance and impedance [28]. Accordingly, high systemic arterial stiffness and pulse pressure are associated with an increased RRI, even in physiological conditions such as aging [87]. Mediated by increased blood pulsatility in a renal vascular bed with low impedance, pulse wave velocity, or central pulse pressure, markers of aortic stiffness and atherosclerosis are strongly correlated with the RRI, independent of intrinsic renal function [28,86,87,88]. Moreover, Calabia et al. determined that an RRI value of more than 0.69 correlates with increased arterial stiffness and atherosclerotic cardiovascular events [28].

Furthermore, in hypertensive patients, the cardio-ankle vascular index, another noninvasive marker of arterial stiffness, correlates directly with the RRI. A value of more than 9.0 is associated with an increased RRI and high cardiovascular risk [89]. In addition, in healthy volunteers, Liu et al. determined that the renal augmented velocity index, a new ultrasound Doppler parameter, has better correlations with pulse pressure, carotid-femoral pulse wave velocity, and intima-media thickness than the RRI [90].

Due to arterial stiffness, the reflected wave returns early in systole and not diastole, thus favoring increased cardiac afterload and left ventricular (LV) hypertrophy [91]. On the one hand, increased pulse wave velocity determines abnormal LV systolic function with a preserved ejection fraction (EF) but decreased longitudinal global strain and high LV twist and, on the other hand, diastolic dysfunction with a normal or high elevated filling pressure [91]. However, central pulse pressure, diastolic trasmitral E and A Doppler flow velocities, and the velocity time integral of the LV outflow tract also have strong correlation with RRI values [23]. Moreover, not only does the RRI depend on LVEF and the myocardial performance index [92], but it is also an independent prognostic marker for atherosclerotic cardiovascular events in patients with preserved EF [93].

In cases of **atherosclerotic stable or unstable coronary artery disease** (CAD) referred for coronary angiography, the RRI is a powerful predictor of death, myocardial infarction, or stroke during the first 24 months of follow-up [94]. A preprocedural RRI value of more than 0.645 associated with a left main lesion correlates with the worst prognosis [94]. However, Doppler-derived renal parameters also have a strong correlation with the extent and severity of CAD [95]. Thus, in acute coronary syndromes, the renal RI and the pulsatility index (PI) are independent predictors of an elevated SYNTAX score [95]. Moreover, RRI is superior to the glomerular filtration rate (GFR) for predicting the worsening of renal function after coronary angiography, because it provides a comprehensive characterization of the hemodynamic and neuro-hormonal factors of cardiorenal syndrome [96]. An RRI value higher than 0.7 predicts renal dysfunction after invasive coronary evaluation with good accuracy [96]. Several mechanisms of worsening renal function due to contrast media are incriminated [96,97]. Increased RRI values and renal vascular resistance favor the occurrence of endothelial dysfunction, cytokine secretion, ischemia, and fibrosis and determine renal vascular rarefaction with the worsening of renal function [96]. Furthermore, contrast media may facilitate an imbalance between vasodilating and vasoconstrictive substances, with the inhibition of nitric oxide synthesis arising due to increased free radicals and reactive oxygen species [96,97]. Whereas viscous contrast media induce direct tubular injury, water-soluble contrast media lead to renal dysfunction secondary to the difference in osmolarity between the arterioles and the interstitial tissue [96,97].

After coronary artery bypass surgery, the incidence of acute kidney injury (AKI) ranges between 15–30% [98]. The postoperative worsening of renal function is associated with prolonged hospitalization and a high risk of cardiovascular complications and death [96,98,99]. The diagnosis of acute renal failure made by serum creatinine and urine output may require up to 48 h, but the RRI has better accuracy in the early detection of AKI [96,99]. Recently, measuring the RRI using Bandyopadhyay method, Kajal et al. found that intraoperative transesophageal echocardiography plays an important role in evaluating not only the cardiac function but also the arterial renal flow at the interlobar or arcuate level [98]. An RRI value greater than 0.7 correlates with renal dysfunction and a value of more than 0.83 predicts the necessity of dialysis [98]. The RRI can be assessed at three crucial moments of the cardiovascular intervention: after the induction of anesthesia and orotracheal intubation, after finishing the cardiopulmonary bypass, and at the end of the surgery. The best predictive value for the occurrence of AKI is the RRI value measured after the surgery is completed [100]. Furthermore, in aortic surgery, the RRI is also superior to serum creatinine and urine output in the early detection of AKI on the first postoperative day [100]. With an accuracy of 76%, the variation between pre- and postoperative RRI values has a net benefit for AKI management of 11% [101].

The renal PI, defined as the difference between systolic and diastolic flow velocities divided by the mean velocity, is another ultrasound parameter that can predict acute renal failure after cardiovascular interventions [102]. Thus, with a cut-off value of 1.86, PI measured at the end of cardiac surgery is a powerful predictor for the development of postoperative AKI with a Youden index of 0.46 [102].

Renal damage due to **arterial hypertension** consists of a reduction in post-glomerular capillaries, chronic ischemia, and sclerosis of the intrarenal arterioles with increased vascular resistance [103]. The diagnosis of subclinical renal damage in hypertensive patients is assessed by GFR and albumin excretion; this combination is an independent predictor of cardiovascular events [103,104]. Moreover, the RRI has been proven to be effective in the early diagnosis and prognosis of renal complications in arterial hypertension [87]. Thus, a high RRI is associated with a faster decrease in renal function, even when the GFR is still normal [86]. In untreated hypertensive patients, the RRI is associated with albuminuria; an RRI greater than 0.7 predicts urinary protein excretion [87]. Increasingly, in hypertensive nephropathy, an increased RRI also correlates with a mild reduction in GFR [87,105].

RRI varies depending on the systolic and diastolic blood pressure (BP). High RRI values corelate with increased morning BP values or variability measurements in outpatients, but not with nocturnal systolic BP [106]. In addition, in a two-year follow-up study with hypertensive patients, Sveceny et al. identified an inverse correlation between the RRI and the 24 h diastolic to systolic BP ratio and the change in pulse pressure [86]. In patients with a GFR lower than 90 mL/min/1.73 m^2^, only the ratio between diastolic and systolic BP values, and not the change in pulse pressure, remains associated with the RRI [86]. Additionally, Kusunoki et al. identified a strong association between the circadian variability of BP and renal dysfunction. Therefore, high systolic and blunted nocturnal BP values correlate with renal parameters (RRI, GFR) and also with high arterial stiffness, as assessed using pulse wave velocity [107].

Furthermore, the RRI is associated with other organ damage due to arterial hypertension, such as the pulse wave velocity and arterial stiffness, LV hypertrophy, carotid artery intima-media thickness, and the retina resistive index of central artery [86,108,109]. Hemodynamic and structural cardiorenal subclinical impairment secondary to hypertension is demonstrated by a powerful correlation between the RRI and the LV mass index, hypertrophy, and diastolic dysfunction [26]. The pattern is also found in hypertensive children, where RRI values are higher compared to those of healthy subjects and are associated with echocardiographic LV parameters: the interventricular septum and posterior wall thickness, the LV mass index, LV EF, and fractional shortening [110]. The RRI also has a strong linear correlation with vascular dysfunction, which is estimated by the carotid intima-media thickness and the total plaque area [111,112]. Moreover, hypertensive retinopathy can be diagnosed early by the ocular resistive index measured at the level of the ophthalmic artery, the central retinal artery, or the posterior ciliary artery; it has good agreement with renal dysfunction, as evaluated by the RRI, GFR, and albuminuria [109,113].

In **heart failure** (HF) patients, regardless of LVEF, the RRI is an important marker of renal dysfunction and cardiovascular outcomes [114]. Thus, a high RRI correlates with reactive oxidative species, endothelial dysfunction, and increased inflammatory cytokine secretion [115]. Moreover, the RRI is influenced by neuro-hormonal activity; this is augmented in HF and depends on the central venous pressure, which is also increased in HF patients [115].

In HF with reduced EF, the RRI correlates with pulse pressure and blood urea nitrogen; meanwhile, in HF with preserved EF, the main predictors for the RRI are GFR and the tricuspid regurgitation peak gradient [114]. In addition, patients with HF with preserved EF and subclinical renal impairment, assessed as having high RRI values, have an increased risk of major cardiovascular events and poor prognosis compared to those with normal renal function [116]. Furthermore, the RRI and the acceleration time measured at the time of hospitalization and after 24 h are powerful predictors of worsening renal function in acute decompensated HF patients, with 89% sensitivity and 70% specificity [117].

In congestive HF, the renal compensatory mechanism linked to increased preload is affected by several factors, such as reduced arterial perfusion, glomerular and tubule-interstitial injuries, and high vein congestion [118]. Currently, the fluid status can be evaluated by noninvasive ultrasound at various sites: the heart, lung, inferior vena cava, or hepatic veins [119]. A renal Doppler ultrasound can assess not only arterial systolic and diastolic velocities and the RRI but also venous parameters, by measuring the intrarenal vein Doppler flow, the venous impedance index, or the venous stasis index [118]. These new markers play an additional role in the diagnosis and evaluation of renal congestion and guide volume management in heart failure or intrinsic renal dysfunction [118]. Recently, Wallbach et al. identified an improvement of the intrarenal venous flow and impedance index in acute decompensated HF patients with LVEF values below 35% at discharge compared to the first 48 h of hospitalization; this was due to maximal medical therapy being used for every patient, according to the current guidelines [119].

The new ESC guideline for acute and chronic HF updated the medical treatment with two new classes of drugs: sacubitril/valsartan and sodium/glucose transport protein 2 (SGLT2) inhibitors [120]. The benefit of sacubitril/valsartan in HF patients with reduced LVEF is mediated not only by cardiac protection (natriuretic and diuretic effects, increased EF, reverse remodeling, improved diastolic function) but also by favorable renal effects, implying an improvement of cardiorenal syndrome [115]. Therefore, due to the inhibition of angiotensin II receptors and neprylisin, in the kidney the combination of sacubitril/valsartan favors natriuresis and diuresis, the dilatation of the afferent arteriole with improved GFR, and arterial renal flow, decreasing the RRI significantly from 0.67 to 0.649 [115]. In addition, sacubitril/valsartan reduces renal fibrosis via the inhibition of neprylisin, an enzyme that causes efferent arteriole dilatation, glomerular hypertrophy, and increased mesangial tissue [115].

SLGT2 inhibitors favor natriuresis and reduce renal glucose reabsorption by increasing its urinary secretion [85]. If they were initially used only for the treatment of type 2 diabetes mellitus, independent of insulin levels, SGLT2 inhibitors are now essential drugs for HF therapy regardless of LVEF values [84,120]. However, these drugs have additional positive effects on arterial stiffness and renal function [84]. SLGT2 inhibitors reduce the RRI by several mechanisms: the inhibition of glucose and sodium reabsorption in the proximal tubule, increasing sodium secretion in the macula densa, and decreasing systemic pulsatility [84]. Thus, Solini et al. identified a significant decrease in the RRI, from 0.62 to 0.59, after only two days of dapagliflozin treatment [84]. Moreover, in a rat histopathological model, treatment with dapagliflozin versus diabetes without SGLT2 inhibitors reduced all inflammatory and apoptotic parameters from the tubular renal cells [121]. The main correlations between and prognostic values of RRI and cardiovascular diseases are listed in Table 4.

### 3.3. RRI Interaction with Other Diseases

**Non-alcoholic fatty liver disease** is a common metabolic disorder with systemic manifestations [122]. Hepatic steatosis and fibrosis correlate with subclinical cardiovascular dysfunction assessed by the LV mass index, diastolic function, pulse wave velocity, and carotid intima-media thickness, and also with renal function as evaluated by GFR and the RRI [122,123]. Compared to healthy subjects, patients with non-alcoholic fatty liver disease have an increased RRI and reduced GFR. Moreover, an RRI greater than 0.62 correlates with a high risk of renal impairment secondary to liver disease [123].

In **paediatric cirrhosis**, the occurrence of kidney damage is associated with a poor prognosis [124]. The early diagnosis of renal impairment becomes essential if liver disease is to be better managed [124,125]. An RRI greater than 0.7 is found in 32% of children with chronic liver disease with or without ascites [124]. However, tense ascites with an RRI value of more than 0.7 at the time of hospitalization is associated with a higher risk of AKI, ascites recurrence, readmission, and mortality [124]. Optimal hepatic therapy with paracentesis and several albumin infusions improves renal function and decreases the RRI [124,125].

**Systemic sclerosis** is an autoimmune disease that affects the internal organs and skin through inflammation, vascular dysfunction, and fibrosis [126]. The vasculopathy in systemic sclerosis includes pulmonary arterial hypertension, peripheral cutaneous artery disease with Raynaud’s phenomenon, ulcers or gangrene, and renal arterial disease [126,127]. The RRI is a noninvasive ultrasound parameter that is able to diagnose early renal impairment in systemic sclerosis, before irreversible structural arterial changes occur [127]. An RRI greater than 0.7 correlates with a longer disease duration and an increased risk for the occurrence of digital ulcers [126]. Moreover, pulmonary hypertension in systemic sclerosis is associated with renal dysfunction [126,127]. A high RRI and a low GFR favor a three-fold increased risk of mortality in these patients [126].

**Juvenile idiopathic arthritis** is one of the most common rheumatic diseases in children [128]. Amyloidosis is the most common renal lesion in juvenile idiopathic arthritis, but membranous glomerulopathy, mesangial nephropathy, focal glomerulosclerosis, and antineutrophil cytoplasmatic antibody-negative glomerulonephritis are also described [128,129]. Albuminuria is the gold standard for the diagnosis of renal injury, but the RRI is a more sensitive parameter that can identify renal dysfunction early in rheumatic arthritis [128]. In addition, RRI has a linear correlation with C-reactive protein and the JADAS (Juvenile Arthritis Disease Activity Score) scale, suggesting an additional inflammatory mechanism of renal injury, associated with endothelial dysfunction and subclinical atherosclerosis [128,129].

**ꞵ-thalassemia** is a chronic anemia characterized by a reduction in or the absence of beta-globin synthesis [130]. This chronic anemia decreases arterial resistance with secondary hyperdynamic circulation and increased GFR [130]. Glomerular hyperfiltration accelerates mesangial sclerosis and tubular damage, which favors proteinuria, hypercalciuria, hyperuricosuria, and the elevated excretion of beta-microglobin [130,131]. Thereby, ꞵ-thalassemia patients have elevated GFR with normal or reduced creatinine serum levels but an increased RRI from the early stages of the hematological disease [131]. In addition, RRI variation and delta RRI, parameters obtained during renal stress tests, are useful tools for the early diagnosis of subclinical renal dysfunction in ꞵ-thalassemia and may improve the management of this disease [130]. The main correlations between and prognostic values of RRI and extracardiac diseases are listed in Table 5.

## 4. Conclusions and Future Prospects

Despite the initial expectations, RRI is not specific for certain causes of kidney dysfunction, nor for CKD and in allografts. In diabetic patients with renal dysfunction, an increased RRI may favor diabetic kidney disease but no other causes of kidney disease. In patients with renal artery stenosis, RRI not on the side of stenosis but in the contralateral kidney seems to better predict renal outcomes after revascularization. Moreover, the RRI may distinguish between reversible and irreversible AKI and indicate the risk of AKI occurrence in different categories of critically ill patients. In patients with transplanted kidneys, the RRI does not correlate with graft prognosis, but it has been shown to be significantly associated with recipient survival, the central pulse pressure, and aortic stiffness parameters, highlighting the important influence that systemic circulation exerts on the RRI. In addition, the RRI is an important marker of renal subclinical dysfunction in different cardiovascular diseases and also in various pathologies. Thus, the RRI constitutes an essential marker for the diagnosis of subclinical renal dysfunction in intrinsic kidney diseases and also in cardiovascular and extracardiac pathologies, with significant prognostic value. In kidney disease, both when used alone and when associated with eGFR or the albuminuria level, the RRI predicts early renal impairment. In cardiovascular or extracardiac various pathologies, the RRI is a sensitive marker of secondary renal dysfunction. In association with other parameters of atherosclerosis, inflammation, or target organ damage, the RRI has additional value in the early diagnosis of kidney disease, readmission, prognosis, and mortality of the systemic disease. This finding opens up a vast field of research, bringing to light the interplay between macrocirculation and intrarenal hemodynamic conditions. Therefore, the RRI should be considered an important marker of cardiovascular risk, beyond its prognostic importance for early the diagnosis of kidney damage.

## Figures and Tables

**Figure 1 diagnostics-13-01687-f001:**
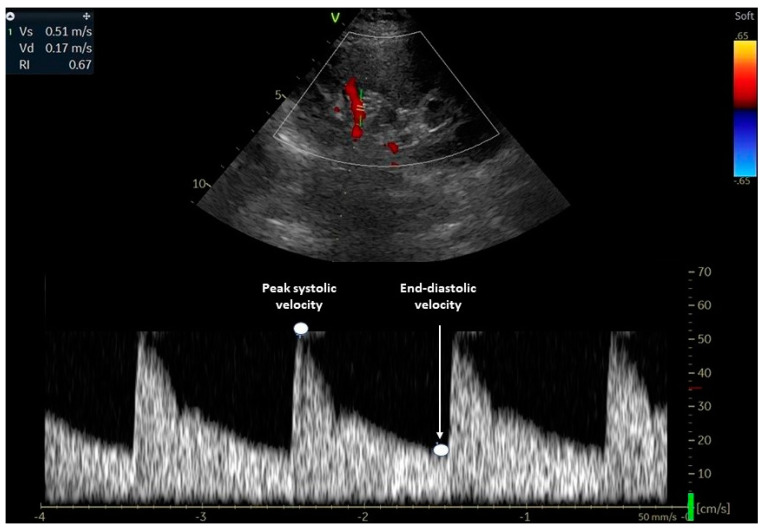
Evaluation of the renal resistive index using Doppler ultrasound. The transducer is placed in an interlobar artery and the spectral Doppler examines the peak systolic and end-diastolic velocities. The renal RI is calculated using the following formula: (peak systolic- end-diastolic)/ peak systolic. RI: resistive index.

**Figure 2 diagnostics-13-01687-f002:**
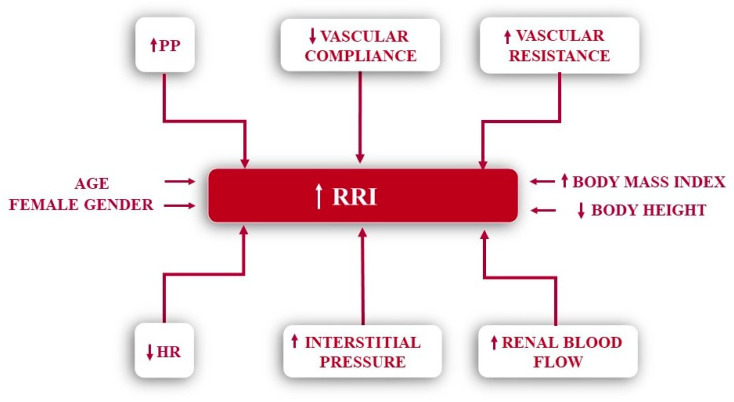
Determinants of the renal resistive index. HR—heart rate; PP—pulse pressure; RRI—renal resistive index.

**Table 1 diagnostics-13-01687-t001:** Standardized requirements for evaluating the renal resistive index with Doppler ultrasound [5,6,7,8,9].

Definition	RRI = (Peak Systolic Velocity − End-Diastolic Velocity)/ Peak Systolic Velocity
**Patient position for examination**	Dorsal decubitus
**Anatomical** **landmarks**	Visualization of the kidney in the longitudinal axis
**Vessels of interest**	Interlobar arteries (adjacent to medullary pyramids)
**Transducer**	Curvilinear low-frequency transducer
**Adjustments for** **image optimization** ■ **Color Doppler** ■ **Pulsed Doppler**	Highest gains possible, avoiding “color bleeding” Lowest filters Low pulse repetition frequency of 1–1.5 kHz while, as far as is possible, limiting aliasing phenomenon Sample volume of 1–2 mm placed in the middle of the lumen for spectral signal recording; highest possible gain without noise
**Number of measurements**	Three to five reproducible waveforms in different areas of each kidney (upper, mid, and lower poles) RRIs from these waveforms are averaged to arrive at the mean RRI values for each kidney
**Normal range**	A value of 0.60 ± 0.01 (mean ± SD) is usually taken as normal; the value of 0.70 is considered the upper normal threshold in adults. It is more accurate to relate the RRI to reference values from the general population, variable according to age and sex.

RRI—renal resistive index.

**Table 2 diagnostics-13-01687-t002:** The main studies that have evaluated the prognostic significance of RRI in CKD.

	Study Design	Cuut-Off Value of RRI	Prognostic Significance
**Chronic kidney disease**			
**Radermacher, J. et al., 2002 [37]**	Multivariate regression analysis for determinants of combined end point: more than 50% decrease in creatinine clearance, ESRD with replacement therapy, or death (n = 162, 3 ± 1.4 year follow up).	0.80	Proteinuria and RRI ≥ 0.80—independent predictors of declining renal function
**Sugiura, T. et al., 2009 [38]**	Cox proportional-hazard analysis for the identification of predictors of worsening renal function defined as a decrease of at least 20 mL/min/1.73 m^2^ in GFR (n = 311, 2-year follow up)	0.70	RRI > 0.70, proteinuria (≥1 g/g creatinine) and high systolic blood pressure (≥140 mmHg) are independent predictors for worsening renal function
**Sugiura, T. et al., 2011 [39]**	Same study design with the previous one (n = 281, 4-year follow up)	0.70	RRI > 0.70, proteinuria (≥1 g/g creatinine), low GFR and high systolic blood pressure (≥140 mmHg) are independent predictors for progression of chronic kidney disease
**Bigé, N. et al., 2012 [40]**	RRI measured 48 h before renal biopsy. Most patients had glomerulonephritis and the mean age was lower than that in other studies. Renal function decline was defined as a decrease in the estimated GFR from baseline of at least 5 mL/min/1.73 m^2^/year or the need for chronic renal replacement therapy (n = 35, 18-month follow up).	0.65	RRI ≥ 0.65 is associated with severe interstitial fibrosis and arteriosclerosis and with renal function decline, independent of the baseline estimated GFR and proteinuria/creatininuria ratio
**Kim, J.H. et al., 2017 [41]**	Retrospective study on patients with moderate renal dysfunction—stage 3 or 4 Progression of renal dysfunction was defined as the doubling of the baseline serum creatinine, >50% decrease in the baseline estimated GFR, or the initiation of renal replacement therapy (n = 118)	RI > 0.79	RRI > 0.79—helpful predictor for the progression of renal dysfunction in this category of patients
**Hanamura, K. et al., 2012 [42]**	Patients with CKD who underwent renal biopsy Worsening of renal function based on a reduction in the estimated GFR with >50% (n = 202, 6 year follow up)	0.70	RRI > 0.70, proteinuria, low estimated GFR at baseline and hypertension were independent risk factors for worsening renal function.
**Toledo, C. et al., 2015 [24]**	Retrospective study (n = 1962, 2.2-year follow up)	0.70	RRI associated with increased non-cardiovascular/non-malignant mortality
**Romano, G. et al., 2022 [43]**	Retrospective study (n = 131, 7.5 year median follow up) Decline in renal function: a serum creatinine level increase of at least 0.5 mg/dL	0.80	RRI ≥ 0.80 associated with a faster increase in serum creatinine levels and each 0.1-unit increament of RRI was an independent determinant of 5-year renal disease progression and mortality. RRI as a single marker showed poor discrimination performance
**Lennartz, C.S. et al., 2016 [45]**	Prospective study (n = 403, 4.4 ± 1.6-year follow up)		Routine duplex examinations among CKD patients did not improved risk prediction for the progression of ESRD beyond a validated equation

CKD—chronic kidney disease; ESRD—end-stage renal disease; GFR—glomerular filtration rate; RRI—renal resistive index.

**Table 3 diagnostics-13-01687-t003:** The main studies that have evaluated the prognostic significance of RRI in AKI.

	Study Design	Values of RRI	RRI Significance
**Differentiation between reverisble and irreversible injuries**			
**Platt, J.F. et al., 1991 [73]**	Cross-sectional study on patients with AKI (n = 91)	0.85 ± 0.6 in acute tubular necrosis vs. 0.67 ± −0.9 in prerenal AKI	An elevated RRI (≥0.75) occurred in 91% of patients with acute tubular necrosis versus 20% in those with prerenal azotemia
**Izumi, M. et al., 2000 [74]**	RRI evaluated relative to the fractional exertion of Na, the renal failure index, and the urinary/serum creatinine ratio	0.80	RRI proved to be equal to other validated factors for differentiating between irreversible and reversible ARI
**Darmon, M. et al., 2011 [75]**	Consecutive patients requiring mechanical ventilation (n = 51)	0.795	RRI was 0.71 in the transient AKI group vs. 0.82 in the persistent AKI group. RRI was better than urinary indices for diagnosing persistent AKI
**Ninet, S. et al., 2015 [76]**	Metanalysis including 9 studies (n = 449)		Increased RRI is a good predictor of AKI
**Fu, Y. et al., 2022 [77]**	Retrospective study on patients in shock with RRI measured in the first 12 h of ICU admission (n = 102)	0.70 ± 0.05 in irreversible vs. 0.66 ± 0.05 in reversible AKI	A clinical prediction model combining serum creatinine and the non-renal SOFA score showed a better prediction ability for non-recovery, and the addition of RRI to this model did not improve its predictive performance
**PREDICTION OF ACUTE KIDNEY INJURYY OCCURENCE**			
**Haitsma, M. et al., 2018 [78]**	Mixed ICU patients with and without shock (n = 99)	0.71 in those who developed AKI vs. 0.65 in the control group	High RRI can be used as an early warning signal for AKI due to its high specificity
**Lerolle, N. et al. 2006 [79]**	Patients with septic shock. RRI evaluated in the first 24 h following vasopressor introduction (n = 35)	0.77 ± 0.08 in those who developed AKI vs. 0.68 ± 0.08 in control group	RRI > 0.74 on day 1
**Schnell, D. et al., 2012 [80]**	Critically ill patients with severe sepsis or polytrauma (n = 58) RRI measured within 12 h of admission	0.80 in patients who developed AKI stage 2 or 3 vs. 0.66 in the control group	In a multivariate analysis comparing the predictive value of RRI, serum and urinary cysteine RRI was the only parameter predictive of AKI on day 3
**Bossard, G. et al., 2011 [81]**	Patients undergoing elective heart surgery with pulmonary bypass with at least one risk factor for AKI (n = 65)	RRI in the postoperative period: 0.79 ± 0.08 in patients who developed AKI vs. 0.68 ± 0.06 in those without AKI	RRI > 0.74 in the postoperative period predicted AKI with high sensitivity and specificity
**Peillex, M. et al., 2020 [82]**	Patients who underwent TAVR for severe aortic stenosis (n = 100)	0.80	RRI > 0.80 at one day after TAVR was a significant predictor of AKI

AKI—acute kidney injury; RRI—renal resistive index; SOFA—the Sequential Organ Failure Assessment; TAVR—transcatheter aortic valve replacement.

**Table 4 diagnostics-13-01687-t004:** Cardiovascular diseases for which the prognostic significance of RRI has been evaluated.

Cardiovascular Disease	Parameters of Cardiovascular Disease	RRI Cut-Off Value	RRI Significance
**Arterial stiffness**	Pulse wave velocity Central pulse pressure Cardio-ankle vascular index	0.69	Increased RRI is a good predictor for arterial stiffness, with no influence from intrinsic renal functions, and it is modified with aging
**Coronary artery disease**	Coronary lesions identified using angiography	0.645 for severity of CAD 0.7 for renal failure	Increased RRI is a powerful predictor for CAD with no discrimination for specific coronary artery lesions Increased RRI predicts renal dysfunction after coronary angiography or aorto-coronary bypass
**Arterial hypertension**	Systolic and diastolic BP Pulse pressure	0.7	Increased RRI correlates with severe arterial hypertension and is a good predictor of renal dysfunction secondary to arterial hypertension
**Heart failure**	LVEF measured using echocardiography	0.649 (sacubitril + valsartan) 0.59 (dapagliflozin)	Increased RRI is a good predictor of renal dysfunction secondary to heart failure A reduction in RRI due to medical therapy for heart failure is associated with good prognosis

CAD—coronary artery disease; LVEF—left ventricular ejection fraction; RRI—renal resistive index.

**Table 5 diagnostics-13-01687-t005:** Other diseases for which the prognostic significance of RRI has been evaluated.

Cardiovascular Disease	Parameters of the Disease	RRI Cut-Off Value	RRI Significance
**Non-alcoholic fatty liver disease**	Hepatic steatosis Hepatic fibrosis	0.62	Increased RRI correlates with early renal dysfunction
**Paediatric cirrhosis**		0.7	Increased RRI correlates with early renal dysfunction and is a good predictor for readmission to hospital and mortality
**Systemic sclerosis**	Pulmonary hypertension Cutaneous ulcers and gangrene	0.7	Increased RRI correlates with early renal dysfunction, before arterial changes occur, and is a good predictor of mortality
**Juvenile idiopathic arthritis**	JADAS score C-reactive protein	-	Increased RRI correlates with subclinical renal impairment
**ꞵ-thalassemia**	Hemoglobin level Beta-microglobin	-	Increased RRI correlates with early renal dysfunction

JADAS—Juvenile Arthritis Disease Activity Score; RRI—renal resistive index.

## Data Availability

Not applicable.

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
