# Peer review of "Current Insights into the Significance of the Renal Resistive Index in Kidney and Cardiovascular Disease"

_diagnostics, 2023, doi:10.3390/diagnostics13101687_

Round 1

Reviewer 1 Report

This review focuses on the usage of renal resistive index on diagnosis in kidney and cardiovascular disease. The authors summarized the theoretical and experimental evidence to demonstrate the significance of renal resistive index and provided the improvements and problems on current studies regarding RRI. Below are the comments on some minor problems in this review.

1.       It is better to have a summary table on the RRI in different kidney disease (3.1), indicating the usage of RRI on diagnosis, RRI cut-off value and problems on prediction.

2.       Similar summary table or graph can be created for section 3.2&3.3 to indicate the association between RRI and different disease conditions.

3.       Different disease conditions and disease parameters have been summarized in the review, what could be the most significant parameter in combination with RRI on the diagnosis? What is the authors’ opinion on predicting pathological changes by using these parameters (including RRI)?

Author Response

This review focuses on the usage of renal resistive index on diagnosis in kidney and cardiovascular disease. The authors summarized the theoretical and experimental evidence to demonstrate the significance of renal resistive index and provided the improvements and problems on current studies regarding RRI. Below are the comments on some minor problems in this review.

  1. It is better to have a summary table on the RRI in different kidney disease (3.1), indicating the usage of RRI on diagnosis, RRI cut-off value and problems on prediction.

The tables were created, as suggested. They are named Table 2 and 3, for chronic and acute kidney disease.

  1. Similar summary table or graph can be created for section 3.2&3.3 to indicate the association between RRI and different disease conditions.

The tables were created, as suggested. They are named Table 4 and 5, for cardiovascular diseases and extracardiac diseases.

  1. Different disease conditions and disease parameters have been summarized in the review, what could be the most significant parameter in combination with RRI on the diagnosis? What is the authors’ opinion on predicting pathological changes by using these parameters (including RRI)?

RRI is an essential marker for diagnosis of subclinical renal dysfunction in intrinsic kidney diseases but also in cardiovascular and extracardiac pathologies, with significant prognostic value. In kidney disease, used alone or associated with eGFR or albuminuria level, RRI predicts early renal impairment. In cardiovascular or extracardiac various pathologies, RRI is a sensitive marker of secondary renal dysfunction. In association with other parameters of atherosclerosis, inflammation or target organ damage, RRI has additional value in early diagnosis of kidney disease, readmission, prognosis and mortality of the systemic disease.   

Reviewer 2 Report

In this manuscript, authors comprehensively reviewed clinical significance of renal resistive index (RRI) determined by Doppler ultrasound in various kidney disorders and cardiovascular diseases. This paper appears to be interesting and easily understood for many readers, and covered recent findings. The reviewer has a few comments to this manuscript, as follows.

1. In the section 3.1 Renal Resistive Index in Kidney Disease and the final Conclusions, the term diabetic nephropathy should be changed to diabetic kidney disease, because the original studies included heterogenous diabetic subjects with renal dysfunction.

2. The reviewer recommends that authors should make new table summarizing the potential clinical applications of RRI measurement in the section 3.1 Renal Resistive Index in Kidney Disease.

3. In the interventional studies using sacubitril/valsartan and dapagliflozin introduced in page 9, changes in RRI values between before and after the medications should be specifically described.

Author Response

In this manuscript, authors comprehensively reviewed clinical significance of renal resistive index (RRI) determined by Doppler ultrasound in various kidney disorders and cardiovascular diseases. This paper appears to be interesting and easily understood for many readers, and covered recent findings. The reviewer has a few comments to this manuscript, as follows.

  1. In the section 3.1 Renal Resistive Index in Kidney Disease and the final Conclusions, the term diabetic nephropathy should be changed to diabetic kidney disease, because the original studies included heterogenous diabetic subjects with renal dysfunction.

The term was modified, as suggested.

  1. The reviewer recommends that authors should make new table summarizing the potential clinical applications of RRI measurement in the section 3.1 Renal Resistive Index in Kidney Disease.

New tables were created, as suggested. They are named table 2 and table 3, for chronic and acute kidney disease.

  1. In the interventional studies using sacubitril/valsartan and dapagliflozin introduced in page 9, changes in RRI values between before and after the medications should be specifically described.

For sacubitril valsartan, RRI modified from 0.67 to 0.649, with statistical significance (p<0.001).

For dapagliflozin, after only two days of treatment, RRI decreased significantly from 0.62 to 0.59 (p<0.04)..

Data was added in the text of the manuscript.